# Improving forest ecosystem functions by optimizing tree species spatial arrangement

Rémy Beugnon [1,2,3] ✉, Georg Albert [1,4,5], Georg Hähn [1,6,7], Wentao Yu [1,5], Sylvia Haider [8], Stephan Hättenschwiler [3], Andréa Davrinche [1,6,9], Benjamin Rosenbaum [1,5], Benoit Gauzens [1,5] & Nico Eisenhauer [1,10]

Reforestation and afforestation programs are promoted as strategies to mitigate rising atmospheric CO2 concentrations and enhance ecosystem services. Planting diverse forests is supposed to foster such benefits, but optimal tree planting techniques, especially regarding species spatial arrangement, are underexplored. Here, using field measurements from the subtropical BEF-China experiment, we simulate tree growth, leaf litterfall, and decomposition, as a function of various spatial arrangements of tree species, from clusters of species to random distributions. Our simulations suggest that increasing tree species spatial heterogeneity in forests composed of eight tree species leads to higher biomass production, more evenly distributed litterfall, increased litter decomposition, and associated nitrogen and carbon cycling. These effects on forest nutrient dynamics are amplified with increasing species richness. Our data show that the spatial arrangement of tree species is a critical component determining biodiversity-ecosystem functioning relationships. Therefore, we suggest the explicit consideration of spatial arrangements when planting trees for reforestation and afforestation projects.

Promoting carbon sequestration in forests has a strong potential to mitigate increasing atmospheric carbon concentration and thereby climate change[1]. In addition to conserving existing forests, re- and afforestation programs have become an increasingly important practice to counterbalance rising carbon emissions[1,2]. The significance of species-rich forests in enhancing and stabilizing forest functioning is particularly relevant in the context of carbon dynamics and sequestration[3–6]. Forests of high tree diversity promote productivity and soil carbon storage[4,7,8], the latter mostly by enhancing litterfall[9–11], thereby linking aboveground production to soil carbon storage[10,12,13].

Ecosystem functions, such as forest productivity or litter decomposition, are usually assessed at the stand level. Nonetheless, underlying processes (e.g., litterfall, competition for light and nutrient uptake) are spatially constrained[14–18], and determined by interactions among neighboring trees[17,19,20]. For instance, the amount of litter falling on the ground decreases with increasing distance from the individual tree producing the litter, and thus its contribution to the forest floor litter layer and its decomposition[14–16]. Consequently, the spatial arrangement of tree species determines potential species interactions and ecosystem processes, and thus, underpins biodiversity effects commonly assessed at the stand level[21,22]. Spatial heterogeneity of tree

[1]German Centre for Integrative Biodiversity Research (iDiv) Halle-Jena-Leipzig, Puschstrasse 4, Leipzig, Germany. [2]Leipzig Institute for Meteorology, Universität Leipzig, Stephanstraße 3, Leipzig, Germany. [3]CEFE, Univ Montpellier, CNRS, EPHE, IRD, Montpellier, France. [4]Department of Forest Nature Conservation, University of Göttingen, Göttingen, Germany. [5]Institute of Biodiversity, Friedrich Schiller University Jena, Jena, Germany. [6]Institute of Biology/Geobotany and Botanical Garden, Martin Luther University Halle-Wittenberg, Am Kirchtor 1, Halle, Germany. [7]University of Bologna, Department of Biological, Geological and Environmental Sciences, Via Irnerio 42, Bologna, Italy. [8]Institute of Ecology, School of Sustainability, Leuphana University of Lüneburg, Universitätsallee 1, Lüneburg, Germany. [9]Research centre for Ecological Change (REC), Organismal and Evolutionary Biology Research Programme, Faculty of Biological and Environmental Sciences PO BOX 65 00014, University of Helsinki, Helsinki, Finland. [10]Institute of Biology, Leipzig University, Puschstrasse 4, Leipzig, Germany. ✉e-mail: remy.beugnon@idiv.de

species should maximize species interactions, leading to more evenly distributed litterfall that may facilitate decomposition[9], thus enhancing soil carbon sequestration according to the microbial efficiency matrix stabilization framework[9] that, however, still remains to be tested in forest ecosystems and over longer time periods[19,23]. While an increasing number of studies concentrate on examining the effects of spatial heterogeneity on biodiversity-ecosystem functioning relationships at the landscape level (e.g.,[24,25]), there is limited consideration given to the potential benefits of within-habitat spatial heterogeneity of different tree species in enhancing ecosystem functions. Yet, in diverse tree communities where forest management practices might target only one or a few species out of many, the spatial arrangement of trees is a challenging but critical aspect for stakeholders and foresters ([26,27], https://www.fao.org/3/I7730EN/i7730en.pdf). Therefore, striking the balance between a favorable taxonomic and spatial composition of species-rich forests and a reasonable impact on the feasibility of forest management is a key component of designing future forests to maximize ecosystem functioning. A growing number of international and local initiatives are focusing on finding combinations from the pool of native species for local planting efforts (e.g., Restor: https://restor.eco/). In addition to this search for optimal species combinations, the effects of tree species arrangements on litter dynamics may provide further insight for effective plantation designs.

Here, we test the effect of tree species spatial heterogeneity on tree biomass, litterfall, and litter decomposition (thereafter, "litter dynamics" refers to litterfall and litter decomposition processes), and how these processes could optimize the tree species diversity effects on ecosystem functioning. First, we simulate forests with eight tree species following gradients of species spatial heterogeneity, ranging from species planted in blocks to fully random designs including designs with lines or mini-blocks of species (Fig. 1). Second, using tree-tree interaction models[22] and litterfall measurements taken from a large forest Biodiversity-Ecosystem Functioning experiment (BEF-China[28], Fig. 1[14]), we simulate tree biomass and fit a distance-based and species-specific litter distribution model to predict the litter distribution of each tree previously simulated. Third, using data from decomposition experiments from the same experiment[14], we fit a litter-specific decomposition model based on litter species composition and the amount of litterfall, and predict litter decomposition for each square-decimeter on the forest floor of our simulated forests. Thus, we estimate the effect of tree species spatial heterogeneity on tree biomass, litterfall distribution, litter decomposition, and carbon and nitrogen dynamics. Finally, we evaluate how an increasing tree species richness (2, 4, and 8 different tree species) interacts with the tree species spatial heterogeneity-litter dynamics relationship by replicating the experiment for each forest species richness level.

We expect the spatial heterogeneity of tree species distribution to drive species-specific tree biomass and litterfall, which in turn affects spatial patterns of forest floor litter accumulation and consequently litter decomposition processes in eight-species mixtures (H1). As heterospecific interactions among trees will increase with higher species richness, we expect increasing tree species richness to strengthen the relationship between tree species spatial heterogeneity and litter dynamics (H2, Fig. 1A).

Overall, our findings highlight that the spatial heterogeneity of tree species within forests significantly influences tree biomass production, litterfall distribution, and decomposition processes, thereby shaping carbon and nitrogen cycling. These effects are especially pronounced in species-rich forests, where the spatial arrangement of species plays a crucial role in ecosystem functioning. A promising compromise between theoretical optimization and technical feasibility may lie in planting lines of species, which maintains management feasibility while substantially enhancing forest functioning compared to block designs. To support the development of sustainable forest management strategies, future research should adopt integrative approaches that combine fundamental and applied perspectives, emphasizing the role of species interactions and their ecological consequences across spatial scales.

## Results

### Simulation of forest spatial heterogeneity

To study the effect of tree spatial heterogeneity on forest tree biomass and litter dynamics (litterfall and litter decomposition), we simulated these three processes across a range of plantation designs using tree growth models[22], litterfall collections, and leaf litter decomposition data from a field experiment[14]. In short, we simulated two-, four-, and eight-species mixture forests from a pool of eight tree species of a Biodiversity-Ecosystem Functioning experiment in subtropical China (BEF-China[28], Fig. 1B, "Methods"). We selected all 56 two-species mixture permutations, and 1000 four-, and 1000 eight-species mixtures permutations in all potential permutations (Fig. 1B, Suppl. Note S1, "Methods"). Each forest from these 2056 species mixture permutations was "planted" (i.e. simulated) in forest stands of 16 individuals by 16 individuals of trees planted at a one-meter distance ($15 \times 15$ m forests, comparable to the experimental site design of the field experiment) with eight different types of spatial distributions from blocks of species to fully random distributions of the species (Fig. 1B, Suppl. Note S1). In addition, mixtures were also planted in mini-blocks, double lines, and single lines of species, to better represent more realistic and feasible plantation design (Fig. 1B, Suppl. Note S1). Taken together, we simulated 22'308 forest stands (i.e., all species mixture permutations across all planting designs). The biomass of individual trees was estimated based on the models developed by Yu et al., which account for species-specific tree-tree interactions[22]. From tree- and species-specific Bayesian models fitted on our experimental litterfall and decomposition data (Fig. 1B, "Methods" and Suppl. Note S2), we simulated the litterfall distribution of each tree and litter decomposition for each $0.1 \times 0.1$ m pixel in the simulated forest (Fig. 1B). We measured the spatial heterogeneity of tree species in a plot as the deviation of observed conspecific neighbors from those predicted by a null model (i.e., a randomly distributed eight-species mixture following a hypergeometric distribution). In short, we calculated how different a plantation design is from a randomly planted eight-species mixture in terms of mono- vs. heterospecific direct neighbors. Planting lines rather than blocks of species increased the tree species spatial heterogeneity, reducing the difference to the random setting by 48% (Suppl. Note S1). However, planting the species in sets of double lines rather than single ones only reduced this spatial heterogeneity difference by 13% (Suppl. Note S1).

### Effects of tree spatial heterogeneity in forest stands composed of eight tree species

In eight-species mixtures, our simulations showed that tree biomass increased with increasing tree spatial heterogeneity (+1.100 g/m² when comparing block and random designs, Fig. 2A). Surprisingly, the amount of litterfall (average amount of litterfall at forest level) remained constant across heterogeneity levels ($54.3 \pm 1.0$ g/m²), yet, the variability between pixels decreased with increasing species spatial heterogeneity (from 31.6 to 16.6 g/m²), showing a homogenization of the litterfall amounts at the forest stand level (Fig. 2B). Likewise, average litter species richness in a given pixel increased with increasing tree species spatial heterogeneity in eight-species mixture forest stands (from 1.73 to 5.75 species per pixel, Fig. 2B).

### Tree spatial heterogeneity increases litter decomposition and stabilizes decomposition in space

Our results for forest stands composed of eight different tree species highlight an increase of decomposition (i.e., carbon and nitrogen loss rate) with increasing tree species spatial heterogeneity (from 36.5 to

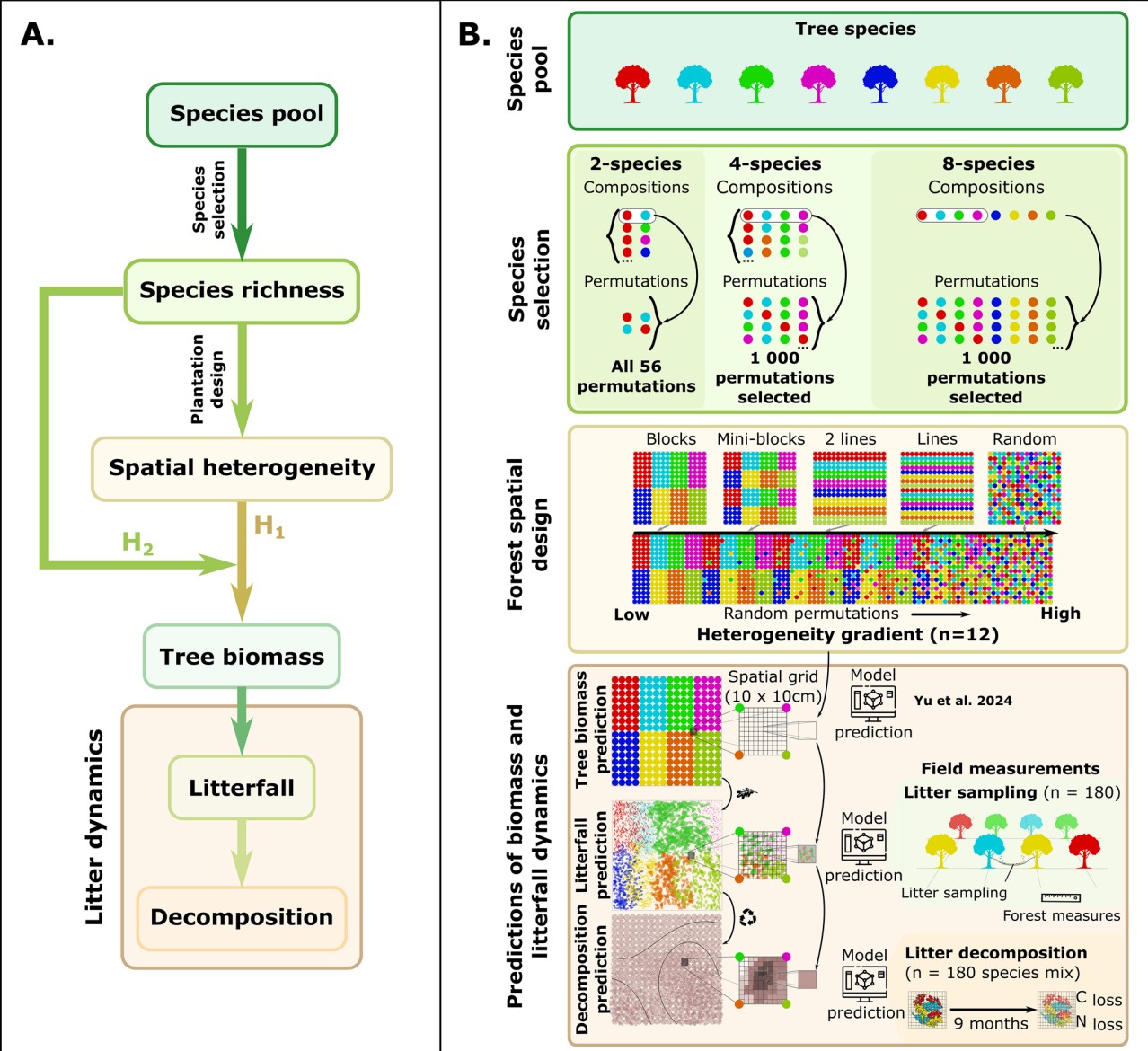

**Fig. 1 | Study hypotheses and the associated experimental design. A** Introduction of the hypotheses linking tree biomass and litter dynamics (i.e., litterfall and decomposition) to species spatial heterogeneity (H1) and the mediation by species richness (H2). **B** Description of the simulation design including species selection, plantation patterns, the prediction of tree biomass based on Yu et al.[22], and the prediction of litterfall dynamics based on field measurements of litterfall and decomposition. Each dot representing a tree is colored according to the species, and each forest stand is composed of 16 by 16 trees.

47.1% for carbon when comparing block and random designs, Fig. 2B, Suppl. Note S3 for additional variables). Similar to the amount of litterfall, we showed a decrease in the spatial variability of litter decomposition (from 7.2 to 5.0%, Fig. 2B), highlighting a spatial stabilization of the decomposition process with increasing tree species spatial heterogeneity. Changes in litter distribution and its consequences for litter decomposition increased average carbon loss at the forest stand level (from 8.41 to 10.30 g/m², Fig. 2C). Moreover, planting the eight species in lines rather than in blocks increased the decomposition rate by 10%, thus reducing the difference observed between the fully random design and block design by almost a third (e.g. for carbon mean decomposition: block design = 36.5%, line design = 40.4%, random design = 47.1%, Fig. 2C). Analyses of nitrogen loss showed similar patterns (decrease of nitrogen decomposition rate by 33%, Suppl. Note S3), highlighting the stimulating effect of tree species spatial heterogeneity on both, carbon recycling and nitrogen release.

## Tree spatial heterogeneity effect strengthens with tree species richness

In order to evaluate how varying spatial heterogeneity modulates tree species richness effects on litterfall and decomposition, we repeated our simulations for two- and four-species mixtures and compared those to the results from the eight-species mixtures (Fig. 3). Tree species richness increased tree biomass (Fig. 3A1), the amount of litterfall (Fig. 3B1), litter species richness (Fig. 3B3), and litter decomposition (Fig. 3C1), as well as, their spatial heterogeneity. Likewise, tree spatial heterogeneity increased litter species richness and decomposition rate but reduced litterfall (Fig. 3B2) and decomposition spatial heterogeneity (Fig. 3C2, black arrows). Across the response variables, we showed a significant interaction effect between tree species richness and spatial heterogeneity ($p < 0.001$), except for tree biomass (Fig. 3A1) and the amount of litterfall (Fig. 3B1). We generally observed an increasing strength of forest stand tree species richness effects on the litterfall and decomposition response variables with increasing

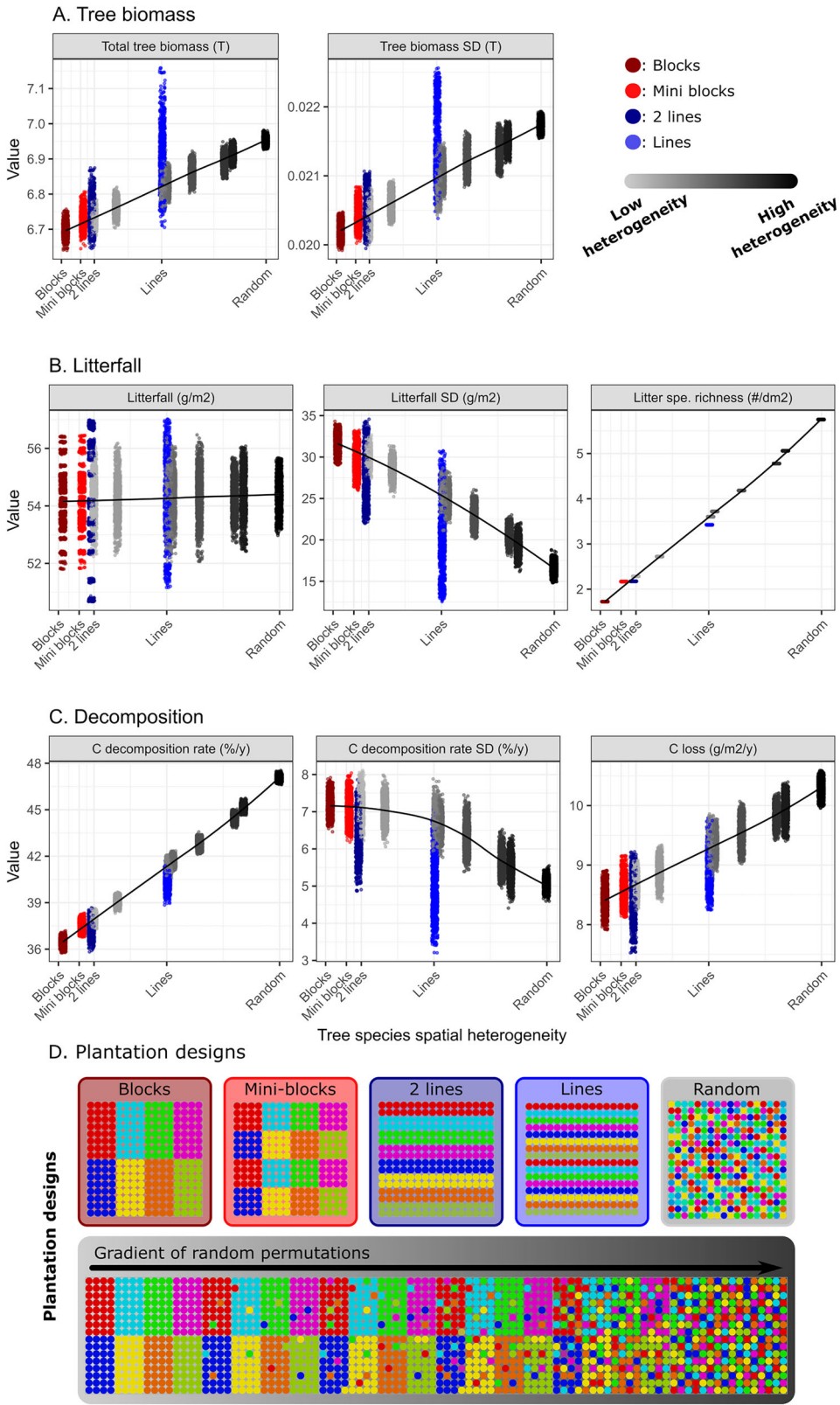

**Fig. 2 | Tree species spatial heterogeneity effect on tree biomass, litterfall, and litter decomposition in 8-species mixtures.** The response variables are grouped in panels: **A** tree biomass, **B** litterfall, and **C** decomposition variables. Tree species spatial heterogeneity (**D**) ranges from blocks of species (dark red) to fully random through the gradient of heterogeneity (gray gradient, Suppl. Note S1), mini-blocks (4 × 4 trees, light red), double lines (dark blue), and single lines of species (blue). Loess regressions fitted on the tree species spatial heterogeneity gradient (in gray, excluding lines and mini-block designs) were added to highlight the patterns.

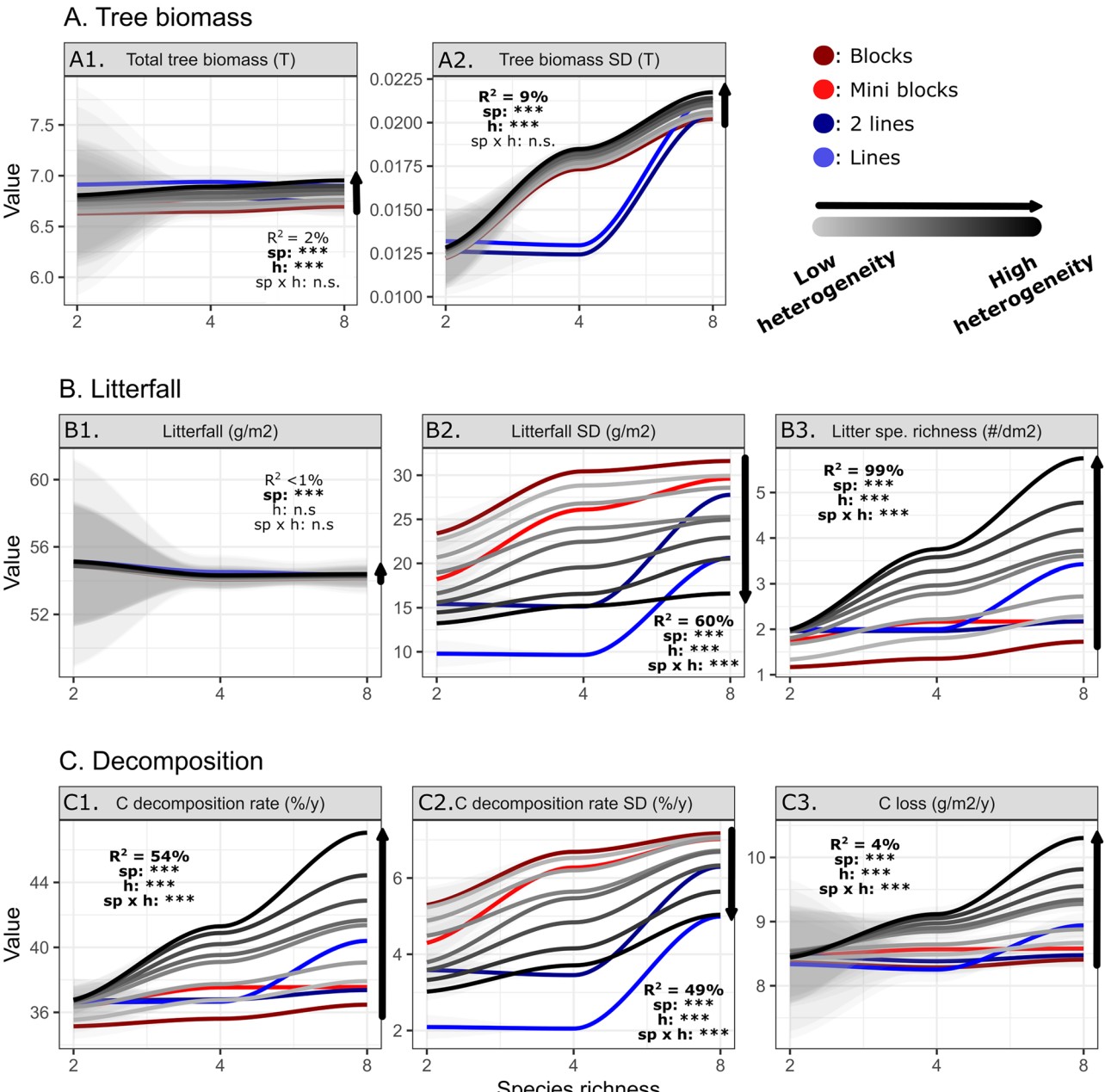

**Fig. 3 | Interactions between tree species richness and tree species spatial heterogeneity.** The interactions were tested on **A** tree biomass, **B** litterfall and **C** litter decomposition using a diversity gradient from two species mixture ($n = 56$) to four ($n = 1000$) and eight-species mixtures ($n = 1000$) and an 8 levels gradient of tree species spatial heterogeneity ranging from a block (in red) to a fully random design (gray color gradient). The loess regression lines were added for each heterogeneity level, and gray ribbons highlight the 95% confidence interval around the mean values. Model ANOVA outputs are reported.

tree species spatial heterogeneity (Fig. 3B, C); however, we did not identify such an effect on tree biomass variables (Fig. 3A). For instance, carbon decomposition rate increased with tree species richness, and this relationship strengthened with increasing tree species spatial heterogeneity ($p_{interaction} < 0.001$ and $R^2 = 54\%$, Fig. 3C1); thus, spatial heterogeneity boosted the effects of diversity on ecosystem functions. For block designs, tree species richness effects on decomposition rate became neglectable ($p > 0.05$, from 35.1 to 36.5% between 2- and 8-species mixtures, Fig. 3C1), while in the random settings, they were more pronounced ($p < 0.001$, from 36.7 to 47.1%). In summary, our results show that the strength of the BEF relationship is contingent upon the spatial heterogeneity of tree species within the forest.

## Discussion

By integrating empirical data from extensive field measurements of litterfall distribution and litter decomposition rate into simulated forests varying in heterogeneity of the spatial arrangement of tree species, we showed that this heterogeneity strongly affects tree bio-mass production, litterfall, and litter decomposition. With increasing spatial heterogeneity, tree biomass increases, the litter layer is more diverse and most importantly, more evenly distributed across the forest floor without changing the total amount of litterfall. These changes boost decomposition rates and thereby enhance the total amount of carbon and nitrogen processed by the ecosystem (H1). The effects of tree species spatial heterogeneity on litterfall spatial het-erogeneity and decomposition are particularly pronounced in species-

rich forests (H2). Our results suggest that the spatial heterogeneity of tree species within forests has a notable impact on the distribution patterns of litterfall, subsequent litter decomposition, and thus carbon and nitrogen dynamics, which should be considered in sustainable forest management.

## The effects of species spatial heterogeneity on tree biomass, litterfall, and decomposition

Forest tree species spatial heterogeneity has direct effects on forest productivity, litterfall, and decomposition. The reduced variability of species-specific litter biomass leads to the spatial homogenization of the litter composition in the forest. The negative correlation between tree spatial heterogeneity and litter spatial variability can be explained by the spatially restricted distribution of litter falling from the canopy (Suppl. Note S2[14,16]). Falling litter from a particular tree tends to stay close to touch the ground in close vicinity to that tree, which means that in a block design regrouping the different species, their litters only mix at the edges of the blocks. Conversely, a spatially heterogeneous design with a more even distribution of the different species throughout the forest allows better litter mixing and a more uniform composition of litter from each species across the entire forest. Processes like wind and topology[29], but also animal litter translocation can additionally lift the litter spread limitation[30], therefore potentially contributing to the positive effects of increasing the spatial distribution of litter from different tree species.

Our findings suggest that positive effects of tree species spatial heterogeneity on litter decomposition rates (H1) result from an increased litter diversity associated with higher spatial homogenization. Litter species richness was shown to promote litter decomposition[31,32], with faster litter decomposition up to the stand-level[14]. Mixing of litter from different species can increase decomposition by promoting nutrient transfer between different types of litter[31,33], or by increasing litter-dwelling detritivore activity by higher resource complementarity[34]. In addition, increasing tree species heterogeneity fosters higher spatial stability in litterfall and litter decomposition processes in space. Decomposition is a key ecosystem process linking carbon and nutrient pools between the atmosphere, the biosphere, and the pedosphere, specifically by linking tree primary production and soil carbon stocks[9,10]. Thus, improving tree productivity and stabilizing litter decomposition by increasing tree species spatial heterogeneity might become a powerful tool to optimize biodiversity benefits for nutrient recycling, carbon sequestration, and climate change mitigation. However, the consequences of increasing tree species spatial heterogeneity on soil carbon sequestration remain to be explored in future long-term experiments.

## Interactions between tree species diversity and spatial heterogeneity

Numerous studies highlight that high species diversity enhances multiple forest ecosystem functions and their stability[35], including those directly relevant to forest management, namely forest productivity and carbon sequestration[4,8,36,37]. Our findings provide evidence that tree species spatial heterogeneity is an additional, and so far largely neglected, component of how tree species diversity can influence ecosystem functioning. Accordingly, the positive relationship between tree species richness and ecosystem functions, such as litterfall distribution and decomposition, is maximized when the different tree species are planted fully randomly and remains limited when the different tree species are aggregated into blocks, i.e. patches of single species (Fig. 3). The interaction between tree species richness and tree species spatial heterogeneity highlights the crucial aspect of species interactions for ecosystem functioning[19,20]. In addition, the modulating impact of tree species spatial heterogeneity on the tree species richness effect may be a key to the understanding of the

variability in the strength of BEF relationships in the literature[38,39]. For instance, based on our results, differences in experimental designs, specifically between completely random designs (e.g. BEF-China[28]) to more clustered designs (e.g. FORBIO experiment[40]) can have substantial effects on response variables. Moreover, most experiments do not include any variations in spatial heterogeneity of tree species, in addition to species diversity. Particularly in situations with limited species diversity, our findings indicate that the influence of tree species richness as such is minimal at low spatial heterogeneity and clearly increases with increasing species spatial heterogeneity. Consequently, as tree species richness decreases, prioritizing the consideration of tree species spatial heterogeneity becomes increasingly decisive for enhancing ecosystem functioning. Taken together, these results highlight the need to integrate the small-scale spatial distribution of species into BEF analyses for a better understanding of processes with strong spatial constraints. Here, the significance of heterogeneity in biodiversity-ecosystem functioning (BEF) relationships appears to be closely tied to the scale at which it is examined[15]. It necessitates not only an awareness of its spatial extent but also a comprehension of the fundamental processes at play. In particular, processes such as decomposition exhibit a strong connection to the dispersal potential of litter and decomposer communities. Consequently, defining the appropriate spatial scale of interest ranges from just a few millimeters to centimetres for nutrient transfers between different litter types[41] to a few meters when considering the spread of decomposers[42].

## Consequences for forest management

When it comes down to the reality of forest management decisions, species-rich forests are often considered too costly and even as not feasible practically, despite the overall beneficial effects of species diversity on forest sustainability (e.g., productivity, stability, nutrient cycling)[6]. Our simulations specifically addressed the question of how different planting designs affect decomposition-related ecosystem processes in order to evaluate the trade-off between the beneficial tree diversity effects and the feasibility of management. We demonstrated that the easiest management solution with a simple block design ranks lowest in terms of biodiversity effects on ecosystem processes. On the other hand, a completely random planting design, maximizing spatial heterogeneity, is the ideal scenario from an ecological perspective, but rendering management difficult. An interesting compromise could be planting trees in lines that can still strongly improve forest functioning compared to block designs, while keeping forest management feasible.

## Conclusions

Climate change and biodiversity loss are listed as major global change drivers of ecosystem functioning[43,44]. Re- and afforestation are identified as nature-based solutions to mitigate climate change by counteracting the rising atmospheric $CO_2$ concentration and safeguarding ecosystem functioning[1,2,45]. By explicitly accounting for tree species diversity, newly planted forests may be particularly effective for climate change mitigation[5,45], and as we show here, the spatial arrangement of different tree species additionally has a marked impact on ecosystem functioning, even at low species numbers. However, the diversification of existing and newly planted forests may be challenging for current management practices, requiring different options for optimal solutions. Our findings provide a clear quantification of different planting solutions for key ecosystem processes, demonstrating that any promotion of species spatial heterogeneity enhances the biodiversity effect on ecosystem functioning. Simply transforming plantations of blocked species clusters into adjacent rows of different species was shown to already modify ecosystem functioning, with probably acceptable consequences for forest management. To develop sustainable management strategies, future research should integrate both fundamental and applied perspectives in experimental

designs, with a specific focus on studying species interactions and their ecosystem consequences across spatial scales.

## Methods

Our study has a hybrid design, where mathematical simulations are informed by field observations from the BEF-China experimental site[28]. From our field experiment, we derived statistical models predicting the amount of litterfall and decomposition rates at a given location depending on surrounding tree diversity and species identity. We used these first results and the models by Yu and colleagues[22] to inform our simulations to investigate how different plantation designs would determine tree biomass, litterfall, and litter decomposition.

### Study site

The study site is located in southeast China near Xingangshan city (Jiangxi Province, 29.08–29.11° N, 117.90–117.93° E). Our experimental site is part of the BEF-China experiment (site A[28]), and it was planted in 2009 after a clearcut of the previous commercial plantation. The region is characterized by a subtropical climate with warm, rainy summers and cool, dry winters with a mean temperature of 16.7 °C and a mean annual rainfall of 1821 mm[46]. Soils in the region are Cambisols and Cambisol derivatives, with Regosol on ridges and crests[47,48]. The natural vegetation consists of species-rich broad-leaved forests dominated by *Cyclobalanopsis glauca*, *Castanopsis eyrei*, *Daphniphyllum oldhamii*, and *Lithocarpus glaber*[28,49].

### Field sampling design

To identify the effect of tree spatial organization on litterfall distribution and decomposition, we measured litterfall and decomposition between 180 pairs of trees with varying neighborhood composition and species richness. Each pair consisted of two neighboring trees (1.28 m), and we defined its neighborhood as the ten trees directly adjacent in the planting grid[19]. Each unique species pair was replicated three times at every tree species richness level (plot species richness of 1, 2, 4, 8, and ≥16 species) whenever possible ("broken stick" design[28]). In total, we surveyed 24 unique combinations of tree species resulting in a total of 180 pairs of trees in 52 plots[14]. Tree mycorrhizal types are known to strongly influence tree performances and ecosystem functioning. Therefore, to test a good balance between monotype (either ectomycorrhizal or arbuscular mycorrhizal type) pairs and mixed ectomycorrhizal–arbuscular mycorrhizal pairs, twelve tree species out of the sixteen existing species were surveyed (Suppl. Table S1).

**Tree measurements and biomass estimations.** Tree biomass, used for fitting litterfall models, was predicted for all tree pairs and their neighbors using tree basal area (BA) and species-specific allometric relationships estimated on the pair of trees. (1) Circumference at breast height (CBH) was measured in September 2018 for all pairs of trees and their direct neighbors in order to calculate the basal area of these trees as $BA = (CBH)^2/4\pi$. (2) Tree height was measured for the pair of trees, and tree biomass was calculated following Huang et al.[8]. Tree pairs' BA and biomass were used to estimate species-specific allometric BA-biomass relationships and predict the tree biomass for all neighboring trees[12].

**Measurements of leaf functional traits.** Leaf functional traits were assessed at the species- and plot-level in September 2018, following Davrinche and Haider[50]. For each species in each plot, several leaf samples were collected, and the reflectance spectra were measured using ASD FieldSpec® 4 Wide-Resolution Spectroradiometer (Malvern Panalytical Ltd., Malvern, United Kingdom). Leaf functional traits were predicted from the reflectance spectra of a calibration dataset of the same species, where both reflectance spectra and leaf functional traits were measured. Leaf chemical contents (carbon: C, nitrogen: N) were measured from dried leaves ground into a fine powder (Mixer Mill 400, Retsch, Haan, Germany). About 5 mg of leaf powder was used to determine C and N content with an elemental analyzer (Vario EL Cube, Elementar, Langenselbold, Germany). The relation between the leaf spectra of the calibration samples and the leaf traits was analyzed in the software Unscrambler X (version 10.1, CAMO Analytics, Oslo, Norway) to predict species- and plot-specific leaf functional traits[50].

### Litterfall sampling

In September 2018, one litter trap of 1 m² was set up at a height of 1 m above the soil surface between each pair of trees[14]. Litter was collected during December 2018 to cover the main litterfall season in the region[51]. To measure litterfall composition, each leaf of the litter trap was sorted and identified to species level. Each species' litter was dried at 40 °C for two days and weighed (± 0.1 g).

### Decomposition measurements

We performed a decomposition experiment between the pair of trees to measure total leaf litter decomposition. Large-mesh litter bags (10 cm × 10 cm) were built using a 5 mm-mesh for the upper part of the bag to provide access to macro-decomposers, and a 0.054 mm-mesh at the bottom to prevent loss of fine leaf litter particles, and filled with 2 g (± 0.01 g) of dried litter according to litter trap species composition (i.e., species-specific biomasses) of the different pairs of trees. Therefore, the litter composition of the litterbags matched exactly the litter composition (i.e., species-specific litter masses) collected in the litter traps of the corresponding pair of trees. The litterbags were installed in December 2018 and covered by a 1 m × 1 m grid to prevent dislocation by heavy rainfalls (1 cm mesh size). In September 2019, i.e., after nine months of decomposition and before the start of litterfall, litterbags were collected, water-cleaned and dried at 40 °C for two days. The residual litter was weighed (± 0.01 g) and milled.

Litter C and N content after decomposition were measured from the residual litter with an elemental analyzer (Vario EL Cube, Elementar, Langenselbold, Germany) and corrected for soil contamination following Beugnon, Eisenhauer et al.[14]. C and N loss rates (%) from the litterbags during the period from December 2018 to September 2019 were calculated.

### Simulations

All simulations, statistical analyses, and data visualizations were performed using R software version 4.3 (R Core Team[52]), and R-scripts are provided to the readers on our Zenodo directory (https://doi.org/10.5281/zenodo.13808826). To study the effect of tree spatial heterogeneity on tree biomass, litterfall, and litter decomposition, we simulated a range of forest designs to predict tree biomass using the models by Yu and colleagues[22], and litterfall and litter decomposition using Bayesian models fitted to our field data for 2-, 4- and 8-species mixtures (Fig. 1, Suppl. Note S1). Specifically, we used the output of the biomass simulation to predict litterfall, which then informed the litter decomposition model. The analyses were limited to the eight species matching the field sampling of litterfall and decomposition, and the species predictable from the models by Yu and colleagues[2].

**Species selection.** From all the possible 2-, 4-, and 8-species mixture combinations, we selected all 56 two-species mixture permutations (e.g., Sp1-Sp2, Sp2-Sp1 …), and randomly chose 1000 four- and 1000 eight-species mixtures permutations, totaling 2028 species mixtures permutations. Permutations of the species arrangement for a given species mixture allow for randomized species interactions (example in Suppl. Note S1). Those species mixture permutations were then "planted" (i.e. simulated) across different planting designs.

**Plantation designs in simulations.** The forests were planted in a 16 by 16 grid of trees following the species selection within a plot area of

225 m$^2$ (i.e., 11,377 individual trees per hectare). Tree species spatial heterogeneity was manipulated by using different plantation designs, with a block design, meaning that all individuals from a given species were planted together, and a fully random spatial arrangement of the species at the two extremes of the spatial heterogeneity gradient (Fig. 1, Suppl. Note S1). To create a gradient of spatial heterogeneity, trees were randomly permuted, and eight designs were selected to create a regularly homogeneously distributed gradient from the block to the random design (Suppl. Note S1). In an attempt to more closely connect theoretical planting design options to a realistic forest management context, we further added specific spatial mini-blocks (e.g., mini-blocks of only 4 x 4 individuals per species in eight-species mixtures), double-lines, and single-lines (i.e., one single line of planted individuals per species, Suppl. Note S1). We measured the spatial heterogeneity of tree species in a plot as the deviation of observed conspecific neighbors from those predicted by a null model (i.e., a randomly distributed 8-species mixture). In a random plantation design, the number of heterospecific neighbors X follows a hypergeometric distribution, for which the expected value is $E[X_i] = \frac{n(K_i)}{T-1}$, where n defines the number of neighboring trees, $K_i$ is the number of trees that are not of the same species as tree $i$, and $T$ is the total number of individual trees in the entire simulated plantation. Then, heterogeneity $H$ was defined as $H = \sum(N(i) - E[X_i])$, where the variable $i$ is iterating over all trees in the plantation, and $N(i)$ is the observed number of conspecific neighbors of tree $i$. For each plantation design, we calculated tree biomass, litterfall, and litter decomposition rates based on models fitted to empirical data.

## Tree biomass, litterfall, and litter decomposition predictions

**Tree biomass prediction.** Tree biomass in the different plantation designs was predicted using Eq. (1) following the models from Yu and colleagues, providing a practical method for forecasting tree biomass under varying spatial arrangements at the same experimental site[22]. Specifically, based on tree growth data collected over a period of 7 years (i.e., 2010–2017) in the BEF-China experiment, Bayesian models of individual tree growth were fitted that integrated a species-specific intrinsic growth and pairwise tree interaction term between a focal tree $i$ and its eight neighboring trees $j$. Using the resulting parameters, we could then predict individual tree growth as:

$$B_{t+1,\,i} - B_{t,\,i} = \beta_{s(i)}B_{t,i}^{\theta} + \sum_{j \in n_i} \alpha_{s(i),s(j)}B_{t,j}^{b} \qquad (1)$$

where $B_{t,i}$ is the biomass of individual tree $i$ at time $t$, and $\beta_{s(i)}$ and $\alpha_{s(i),s(j)}$ are species-specific coefficients scaling the intrinsic growth and the pairwise tree interaction, respectively. To allow sublinear effects predicted by metabolic theory, we included allometric scaling exponents $\theta$ and $b$. Using this model fit, individual tree biomass was predicted for each simulated forest (i.e., species composition and spatial design); in particular, we set starting biomasses for all trees to 100 and simulated tree growth for 10 years for all the trees.

**Litterfall model.** To predict the litterfall within our simulated forests, we fitted litterfall models on empirical data, assuming that the leaf litterfall mass $L_i$ of species $i$ at any place within the forest is a function of the tree aboveground biomass and of the distance of the surrounding trees of the same species. We predicted litter biomass $L_i$ accordingly as:

$$L_i = b_{1i}\sum_{j=1...n}B_{ij} + b_{2i}\sum_{j=1...n}d_{ij}^{-1} + b_{3i}\sum_{j=1...n}B_{ij}d_{ij}^{-1} \qquad (2)$$

where $b_{1i}$, $b_{2i}$ and $b_{3i}$ are species-specific coefficients capturing the influence of (1) the biomass of the n surrounding trees of species $i$, $B_{ij}$, (2) the distance to n trees of species $i$, $d_{ij}$, and (3) the interactive effect of biomass and distance. For the model, we considered the 12 trees surrounding the litter traps (i.e., $n \leq 12$; Suppl. Note S2).

**Decomposition model.** We fitted litter decomposition models using the diversity-interaction modeling framework[53]. The model distinguishes between species identity and species diversity effects on litter decomposition. Because any potential non-additive litter diversity effects are difficult to predict, we assumed that diversity effects were purely additive. Hence, litter decomposition was described as:

$$D = \sum_i \beta_i P_i + \sum_{i,j}(\alpha_i + \alpha_j)P_iP_j + b_4L + b_5S \qquad (3)$$

where $\beta_i$ and $\alpha_i$ are species-specific coefficients capturing identity and diversity effects, respectively, with $P_i$ being the proportion of litter mass from species $i$ in the total litter mixture mass $L$. The coefficients $b_4$ and $b_5$ capture the effects of total litter mixture mass, $L$, and litter species richness, $S$.

The litterfall and decomposition models were fitted with Rstan[54], using four Markov chains and 3000 iterations (1000 as warm-up). Model fit, posterior distribution, and quality indices are provided in Suppl. Note S2.

**Predicting litterfall and decomposition in simulated forests.** Using the mean coefficients retrieved from our litterfall model, we predicted litter mass for every species at any given 10 × 10 cm pixel within our simulated forests. This allowed us to determine species-specific litter mass and species richness for each pixel. This was then used to predict litter decomposition rates using the mean coefficients from the decomposition model.

## Reporting summary
Further information on research design is available in the Nature Portfolio Reporting Summary linked to this article.

## Data availability
The data used and generated in this study have been deposited in the Zenodo database under accession code https://doi.org/10.5281/zenodo.13808826.

## Code availability
The RScripts used in this study have been deposited in the Zenodo database under accession code https://doi.org/10.5281/zenodo.13808826.

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

## Acknowledgments

This work was supported by the Deutsche Forschungsgemeinschaft (DFG, German Research Foundation—319936945/GRK2324; 452861007/FOR5281). We gratefully acknowledge the support by the German Center for Integrative Biodiversity Research (iDiv) funded by the German Research Foundation (DFG—FZT 118, 202548816). We thank the TreeDì and Experimental Interaction Ecology research groups for their support, especially the many local helpers for their help with the field sampling. N.E. acknowledges funding by the DFG (Ei 862/29-1, Ei 862/31–1). GA was supported by the MultiTroph Research Unit funded by the German Research Foundation (DFG, 452861007/FOR 5281). The scientific results have (in part) been computed at the High-Performance Computing (HPC) Cluster EVE, a joint effort of both the Helmholtz Center for Environmental Research—UFZ and the German Center for Integrative Biodiversity Research (iDiv). We would like to thank the administration and support staff of EVE who keep the system running and support us with our scientific computing needs: Thomas Schnicke, Ben Langenberg, Guido Schramm, Toni Harzendorf, Tom Strempel and Lisa Schurack from the UFZ, and Christian Krause from iDiv. Supported by the Open Access Publication Fund of Leipzig University.

## Author contributions

R.B., N.E., St.H.: designed the study; R.B., G.H., A.D., Sy.H.: collected field data; R.B., G.A., B.R., B.G.: designed the simulations; R.B., G.A., G.H.: analyzed model outputs; R.B., G.A.: prepared the first draft, R.B., G.A., G.H., A.D., Sy.H., B.R., B.G., N.E., St.H.: revised the manuscript. To promote equity between authors, positions were randomized with the exception for first, second, and senior authors.

## Funding

## Competing interests

The authors declare no competing interests.
