## [Transparent Peer Review file · Nature Communications]

Improving forest ecosystem functions by optimizing tree species spatial arrangement

Corresponding Author: Dr Rémy Beugnon

Version 0:

Reviewer comments:

Reviewer #1

(Remarks to the Author)

The manuscript by Beugnon et al. aimed to understand the effect of tree species spatial heterogeneity on leaf litter dynamics (litterfall production and litter decomposition), and how the species diversity and functional composition of forests affect the relationship between species spatial heterogeneity and litter dynamics. Their simulation study based on the empirical data from the BEF-China are well designed, and clearly showed how the species spatial heterogeneity affected the litterfall distribution and litter decomposition rate, and how the species richness, functional diversity and identity modified the effect of species spatial heterogeneity. The spatial heterogeneity reduced the variability of litterfall distribution and increased the decomposition rate, carbon loss, and nitrogen cycling. The strength of the BEF relationship between tree species richness and decomposition rate diminished as spatial heterogeneity decreased. The study is well designed, their methods are sound, and the manuscript is generally well written. Their results should improve our understanding on how the species spatial heterogeneity affect the relationship between biodiversity and ecosystem function, which should also provide important insight to re-/afforestation management as the authors said. The only concern I have is that the authors claimed that the spatial heterogeneity “improves” ecosystem functioning. The current study showed that the spatial heterogeneity promotes litter decomposition rates and associated carbon losses and nutrient cycling, but did not alter the total amount of litterfall. As carbon storage, particularly those in soils and litter layer are controlled by the balance between litter input and decomposition dynamics, it is not clear from the results of present study alone whether the spatial heterogeneity “improves” carbon sequestration as an ecosystem function. As to the nitrogen cycling, one may say that the spatial heterogeneity improves ecosystem function. Although it was stated in the introduction that soil carbon storage is increased by enhanced litterfall and litter decomposition, no such evidence was found at least in the papers cited there. I totally agreed that the species spatial heterogeneity increased litter decomposition rate, but if it is claimed that this improves carbon sequestration, the logic needs to be more carefully explained or otherwise their claim needs be written in different way. In addition, I listed some issues that should be addressed below. I hope these comments will improve the manuscript.

Additional comments:

1. Line 127: In the panel of Forest spatial design in Figure1 B, it may be better to indicate which design has “low” and “high” heterogeneity for easier understanding although the arrow shows the direction.
2. Line 131: How do you select 12 species out of 24 species planted in BEF-China for litterfall sampling and the decomposition experiment? Are they randomly chosen or chosen based on their leaf traits? How much do they differ in their traits studied? Brief explanation is needed in the “Species selection” of Methods.
3. Line 176: “small” blocks should be “mini” blocks to use same word for figures and text. The order of “small blocks”, “double lines”, and “single line” in the text are better to be ordered along the heterogeneity gradient (i.e, double lines, mini blocks, and single lines).
4. Lines 214 and 216: Please insert “Fig.3C” in the parentheses after “9-species mixture” and “on average”.
5. Line 247: What are CWM1 and CWM2? Which traits related to PC1 and PC2? Need short explanation for them in the main text.
6. Line 249: Leaf toughness seemed not to be measured in this study.
7. In the “Litterfall and decomposition predictions” of the Methods, the equations for litterfall model (equation (1)) and decomposition model (equation (2)) seems missing.

Reviewer #2

(Remarks to the Author)

This study investigates the impact of spatial arrangement on litterfall, litter decomposition and nutrient cycling in the Chinese Biodiversity and Ecosystem Functioning (BEF) tree experiment, building on the findings from Beugnon et al. 2023. The study employs simulations based on experiment-derived coefficients, exploring 2, 4, and 9 species systems across simulated gradients of spatial heterogeneity. For the 9 species system, four common tree planting arrangements are considered: clustered, 1-line, 2-lines, and random.

The study presents important findings for both understanding spatial biodiversity effects on ecosystem processes and implementing diverse tree-planting designs. However, the study's complexity, particularly in methodology, requires careful attention. While diagrams aid comprehension, the presentation of the methodology still lacks clarity. More specifically, the design of the field experiment, from where the parameters are estimated is not thoroughly presented in this paper. Also, the definition (and range of values) of spatial heterogeneity ("the distance from the random distribution of species in terms of average heterospecific direct neighbors") is not clear.

My main concern regards the reliance on simulations, with certain assumptions potentially undermining underlying mechanisms. Specifically, the use of average species-specific biomass in estimating tree biomass overlooks (averages out) tree-tree interactions and spatial arrangement impacts on this key variable. Because litterfall modeling and consequently estimation of decomposition rates relies on tree biomass and conspecific distance, the influence of heterospecific interactions is likely not fully captured. Furthermore, it is not clear how the tree pair identities and interactions with surrounding trees beyond the TSP's are included in the simulations and how they affect the results.

The methodological presentation needs refinement for the broader audience of this journal, addressing concerns about clarity and assumptions. Furthermore, the study's focus on litterfall and decomposition under the carbon sequestration context requires clarification. Carbon sequestration depends on the net balance between inputs and outputs, not just decomposition rates, a nuance overlooked in the study.

Additionally (a minor issue), clarification on the seasonality of litterfall measurements and whether all species shed leaves simultaneously (in December) would enhance the study's context and interpretation.

In conclusion, while the study offers valuable insights, improvements in methodological clarity, addressing oversimplified assumptions, and bridging gaps between experimental and simulated conditions are necessary for a more robust and widely applicable contribution to the field.

Version 2:

Reviewer comments:

Reviewer #1

(Remarks to the Author)

I found the authors have addressed the concerns from my previous review reports. I think the new analyses conducted on tree biomass strengthen the conclusion of this study. This study can notify researchers of the importance of considering the spatial heterogeneity in the BEF studies. One thing I'm newly concerned about is the description of the results related to Fig. 3. I think the authors can say that the effects of species richness on response variables were affected by spatial heterogeneity only when the interaction between tree species richness and tree species spatial heterogeneity was significant. The description of the results related to Fig. 3 was sometimes imprecise. So please describe those results more accurately. In addition, I have listed some minor points that should be addressed below. I hope these comments will improve the manuscript.

Line 107: "litter composition" may be better as "litter species composition".

Line 157 & 159: "lows" should be "lines"

Line 165: Please insert (Fig. 2A) after "designs)".

Line 168: Please insert (Fig. 2B) after "forest stand level".

Line 171: "2A" should be "2B".

In the Figure 2, in the illustration of "Plantation designs", the order of "mini-blocks" and "double lines" should be replaced (i.e., the order should be Blocks, Mini-blocks, 2 lines, lines, and Random from left to right).

Line 191 & 194: "2B" should be "2C".

Lines 202-204: I think this sentence would be somewhat misleading. You can say that tree species heterogeneity strengthens the effects of forest stand tree species richness on the response variables when the interaction between species richness (s) and tree species spatial heterogeneity (h) is significant. Among the 8 response variables in Fig. 3, 5 response variables show the significant interactions, but the variables related to tree biomass and litterfall, which are two of the most important variables in this study, show no significant interactions. Please describe the results more precisely.

Lines 211-212: Same comments as above. Is it only true for C decomposition rate?

Line 225: It's better to insert "distribution" after "litterfall", and "rate" after "litter decomposition".

Lines 229-230: This is not true for tree biomass at least?

Lines 268-271: I agree with this on decomposition rate, but not on biomass production.

Methods

Line 80: "litterfall and litter decomposition" should be "tree biomass, litterfall and litter decomposition"?

Line 133: I was not able to find Suppl S4.

Suppl. S1

Page 1: In "Plot designs", "9-species mixture" should be "8-species mixture".

Page 6: "9-species mixture" should be "8-species mixture".

Page 6-8: "Small blocks" should be "Mini blocks".

(Remarks on code availability)

Reviewer #2

(Remarks to the Author)

Response to authors

I have reviewed the revised manuscript and find that the authors have effectively addressed most of my previous concerns. However, I have a few additional and minor comments that I believe could further clarify and strengthen the paper.

Incorporation of Biomass Estimates

I appreciate the authors' effort to integrate biomass estimates based on the methods from Yu et al. (2024). This addition enhances the analysis and is a valuable improvement to the manuscript. However, from the description in the Methods section, I am uncertain whether these modeled biomass values were integrated into the litterfall and decomposition simulations. From my understanding, biomass should ideally be recalculated in each simulation based on the specific pairwise interaction coefficients relevant to the species composition. This interpretation aligns with the main text and also seems logically consistent with the model framework. However, the methods section does not make this process fully clear. If biomass recalculations were indeed incorporated for each simulation, a brief mention in the methods section would clarify this point. A single sentence where relevant would be enough.

Framing of Soil Carbon Storage

I still have some reservations about the framing of the study's introduction in terms of "soil carbon storage." The current study does not directly measure or model soil carbon storage. While tree biomass, litterfall, and decomposition processes are evaluated and the study concludes an increase in carbon storage within tree biomass, the impacts on soil carbon storage are not explicitly assessed here. Although it is reasonable to hypothesize a relationship with soil carbon (supported by existing literature), the decomposition and mass loss measures here do not directly inform the ultimate fate of carbon (e.g., storage vs. respiration).

Definition of Spatial Heterogeneity

The definition of spatial heterogeneity still needs refinement to ensure replicability. Currently, it lacks sufficient detail for readers to fully replicate or understand its application in the study.

Further, minor comments are provided in the attached files.

(Remarks on code availability)

made.

Point-by-point response to reviewers' comments

Dear reviewers, we would like to thank you for your time and positive feedback on our manuscript. We are grateful for the detailed and constructive suggestions provided that helped us to further improve our manuscript. Please find below a point-by-point response on how we addressed your comments. Please note that we have highlighted all reviewer comments, in *blue* font for the purpose of clarity. Line numbers cited below refer to the “Track changes” document line numbers.

REVIEWER COMMENTS

Reviewer #1 (Remarks to the Author):

The manuscript by Beugnon et al. aimed to understand the effect of tree species spatial heterogeneity on leaf litter dynamics (litterfall production and litter decomposition), and how the species diversity and functional composition of forests affect the relationship between species spatial heterogeneity and litter dynamics. Their simulation study based on the empirical data from the BEF-China are well designed, and clearly showed how the species spatial heterogeneity affected the litterfall distribution and litter decomposition rate, and how the species richness, functional diversity and identity modified the effect of species spatial heterogeneity. The spatial heterogeneity reduced the variability of litterfall distribution and increased the decomposition rate, carbon loss, and nitrogen cycling. The strength of the BEF relationship between tree species richness and decomposition rate diminished as spatial heterogeneity decreased. The study is well designed, their methods are sound, and the manuscript is generally well written. Their results should improve our understanding on how the species spatial heterogeneity affect the relationship between biodiversity and ecosystem function, which should also provide important insight to re-/afforestation management as the authors said.

Response: Thank you for your positive feedback. We greatly appreciate the time you took to review our study.

The only concern I have is that the authors claimed that the spatial heterogeneity “improves” ecosystem functioning. The current study showed that the spatial heterogeneity promotes litter decomposition rates and associated carbon losses and nutrient cycling, but did not alter the total amount of litterfall. As carbon storage, particularly those in soils and litter layer are controlled by the balance between litter input and decomposition dynamics, it is not clear from the results of present study alone whether the spatial heterogeneity “improves” carbon sequestration as an ecosystem function. As to the nitrogen cycling, one may say that the spatial heterogeneity improves ecosystem function. Although it was stated in the introduction that soil carbon storage is increased by enhanced litterfall and litter decomposition, no such evidence was found at least in the papers cited there. I totally agreed that the species spatial heterogeneity increased litter decomposition rate, but if it is claimed that this improves

carbon sequestration, the logic needs to be more carefully explained or otherwise their claim needs be written in different way.

Response: We concur with the reviewer's observation that soil carbon sequestration stands in the balance between soil carbon inputs and outputs, positioning litter decomposition at a critical position, by controlling carbon inputs. Previous studies from the same experiments have shown that tree species richness increased productivity litterfall, especially, neighborhood species-richness (Beugnon, Bu et al. 2023; Ray et al. 2023; Liu et al 2018), with consequences for soil carbon concentrations (Beugnon, Bu et al 2023; Liu et al. 2018). This positive effect of neighboring diversity on forest productivity has recently been linked to positive tree-tree interactions between heterospecific species (Yu et al 2024). Therefore, increasing tree species spatial heterogeneity within the forest is expected to increase species-specific biomass, and thereby decomposition and carbon sequestration.

Tree growth, amount of litterfall, and decomposition are intimately linked. On the one hand, tree productivity enhances the amount of litterfall and litter decomposition (Beugnon, Bu et al. 2023; Beugnon, Eisenhauer et al. 2023; Liu et al. 2018); on the other hand, increasing the amount of litterfall and decomposition enhances nutrient availability in soil, and thereby tree growth (Freschet et al. 2013). Therefore, integrating tree biomass into the simulation might lead to a certain circularity by indirectly integrating plant-soil feedback mechanisms. To tackle this issue, temporal data and models would be needed. We added explanatory text accordingly to the Discussion. Our approach first focuses on the relationship between the spatial organization of tree species and litterfall dynamics. We therefore reduced interacting factors potentially blurring or inflating our results, by keeping covariates, such as tree biomass, constant.

L. 325

“Soil carbon storage depends on the decomposition and the amount of litter set to decompose⁹⁻¹¹. We fixed tree species biomass across the simulation scenarios; and thereby, the amount of litter falling on the ground at the forest level (Fig. 2), to avoid any potential nutrient feedback loops³⁵ and an overestimation of the explored effects. However, recent analyses from the same experiment showed that tree pairwise interactions underlie the biodiversity-ecosystem functioning relationship³⁶. In particular, tree productivity increases with an increasing proportion of inter-specific neighbours, by enhancing tree species spatial heterogeneity, i.e. an enhanced proportion of inter-specific interactions, increases tree productivity and biomass. Taking together Yu and colleagues³⁶ as well as our results highlights that increasing the spatial heterogeneity of tree species increases forest productivity, litterfall and litter decomposition for a given species richness of forest stand. Thus, improving tree productivity and stabilizing litter decomposition by increasing tree species spatial heterogeneity might become a powerful tool to optimize biodiversity benefits for nutrient recycling, carbon sequestration, and climate change mitigation.”

Beugnon, R., Bu, W., Bruelheide, H., Davrinche, A., Du, J., Haider, S., Kunz, M., von Oheimb, G., Perles-Garcia, M. D., Saadani, M., Scholten, T., Seitz, S., Singavarapu, B., Trogisch, S., Wang, Y., Wubet, T., Xue, K., Yang, B., Cesarz, S., & Eisenhauer, N. (2023). Abiotic and biotic drivers of tree trait effects on soil microbial biomass and soil carbon concentration. *Ecological Monographs*, 93(2), e1563. <https://doi.org/10.1002/ecm.1563>

Freschet, G. T., Cornwell, W. K., Wardle, D. A., Elumeeva, T. G., Liu, W., Jackson, B. G., Onipchenko, V. G., Soudzilovskaia, N. A., Tao, J., & Cornelissen, J. H. C. (2013). Linking litter decomposition of above- and below-ground organs to plant–soil feedbacks worldwide. *Journal of Ecology*, 101(4), 943–952. <https://doi.org/10.1111/1365-2745.12092>

Liu, X., Trogisch, S., He, J.-S., Niklaus, P. A., Bruelheide, H., Tang, Z., Erfmeier, A., Scherer-Lorenzen, M., Pietsch, K. A., Yang, B., Kühn, P., Scholten, T., Huang, Y., Wang, C., Staab, M., Leppert, K. N., Wirth, C., Schmid, B., & Ma, K. (2018). Tree species richness increases ecosystem carbon storage in subtropical forests. *Proceedings. Biological Sciences*, 285(1885). <https://doi.org/10.1098/rspb.2018.1240>

Ray, T., Delory, B. M., Beugnon, R., Bruelheide, H., Cesarz, S., Eisenhauer, N., Ferlian, O., Quosh, J., Von Oheimb, G., & Fichtner, A. (2023). Tree diversity increases productivity through enhancing structural complexity across mycorrhizal types. *Science Advances*, 9(40), eadi2362. <https://doi.org/10.1126/sciadv.adi2362>

Yu, W., Albert, G., Rosenbaum, B., Schnabel, F., Bruelheide, H., Connolly, J., Härdtle, W., Von Oheimb, G., Trogisch, S., Rüger, N., & Brose, U. (2024). Systematic distributions of interaction strengths across tree interaction networks yield positive diversity–productivity relationships. *Ecology Letters*, 27(1), e14338. <https://doi.org/10.1111/ele.14338>

In addition, I listed some issues that should be addressed below. I hope these comments will improve the manuscript.

Additional comments:

- 1. Line 127: In the panel of Forest spatial design in Figure 1 B, it may be better to indicate which design has “low” and “high” heterogeneity for easier understanding although the arrow shows the direction.*

Response: Thank you for noticing this, the figure was revised accordingly.

- 2. Line 131: How do you select 12 species out of 24 species planted in BEF-China for litterfall sampling and the decomposition experiment? Are they randomly chosen or chosen based on their leaf traits? How much do they differ in their traits studied? Brief explanation is needed in the “Species selection” of Methods.*

Response: Agreed. The species selection was now detailed in the Methods, and Supplementary Table S1 was added to the manuscript to clarify the designs.

- 3. Line 176: “small” blocks should be “mini” blocks to use same word for figures and text. The order of “small blocks”, “double lines”, and “single line” in the text are better to be ordered along the heterogeneity gradient (i.e, double lines, mini blocks, and single lines).*

Response: Thank you for noticing this; the figure and text were corrected accordingly.

4. *Lines 214 and 216: Please insert “Fig.3C” in the parentheses after “9-species mixture” and “on average”.*

Response: Thank you for noticing this, the figure was corrected accordingly.

5. *Line 247: What are CWM1 and CWM2? Which traits related to PC1 and PC2? Need short explanation for them in the main text.*

Response: We agree that the link between the PCA, and CWMs wasn't clear. Accordingly, we clarified this throughout the revised manuscript.

L. 260 “In addition, both functional axes (CWM 1 and CWM 2) of the community functional identity (corresponding to the two first axes of the leaf functional traits principal component analysis, Suppl. S4) significantly mediated the heterogeneity-litter dynamics relationship (i.e. variable importance above the 0.25 threshold, Fig. 4C, Suppl. S4). For example, a high CWM 1 value, i.e. a low leaf tissue density (depicted by the LDMC) and high nutrient content, led to an augmentation in decomposition rate (Suppl. S3)”

6. *Line 249: Leaf toughness seemed not to be measured in this study.*

Response: Correct. We did not measure leaf toughness but used LDMC as a proxy of the leaf structure. We reworded the sentence to avoid any confusion.

L. 266 “For example, a high CWM 1 value, i.e. a low leaf tissue density (depicted by the LDMC) and high nutrient content, led to an augmentation in decomposition rate (Suppl. S3).”

7. *In the “Litterfall and decomposition predictions” of the Methods, the equations for litterfall model (equation (1)) and decomposition model (equation (2)) seems missing.*

Response: We have to apologize: the equations were lost during the compilation. We added them again.

Reviewer #2 (Remarks to the Author):

This study investigates the impact of spatial arrangement on litterfall, litter decomposition and nutrient cycling in the Chinese Biodiversity and Ecosystem Functioning (BEF) tree experiment, building on the findings from Beugnon et al. 2023. The study employs simulations based on experiment-derived coefficients, exploring 2, 4, and 9 species systems

across simulated gradients of spatial heterogeneity. For the 9 species system, four common tree planting arrangements are considered: clustered, 1-line, 2-lines, and random.

The study presents important findings for both understanding spatial biodiversity effects on ecosystem processes and implementing diverse tree-planting designs.

Response: Thank you for your positive feedback. We greatly appreciate the time you took to review our study.

However, the study's complexity, particularly in methodology, requires careful attention. While diagrams aid comprehension, the presentation of the methodology still lacks clarity. More specifically, the design of the field experiment, from where the parameters are estimated is not thoroughly presented in this paper. Also, the definition (and range of values) of spatial heterogeneity ("the distance from the random distribution of species in terms of average heterospecific direct neighbors") is not clear.

Response: To address this important comment, we revised the entire Methods section to increase clarity. Importantly, we now start our method section with a header paragraph providing a general overview of our approach.

L. 2 (Method)

“Our study has a hybrid design, where mathematical simulations are informed by field observations from BEF-China experimental site1. From our field experiment, we derived statistical models predicting the amount of litterfall and decomposition rates at a given location depending on surrounding tree diversity and species identity. We used this first result to inform our model simulating how different plantation designs would determine litterfall and litter decomposition.”

My main concern regards the reliance on simulations, with certain assumptions potentially undermining underlying mechanisms. Specifically, the use of average species-specific biomass in estimating tree biomass overlooks (averages out) tree-tree interactions and spatial arrangement impacts on this key variable. Because litterfall modeling and consequently estimation of decomposition rates relies on tree biomass and conspecific distance, the influence of heterospecific interactions is likely not fully captured.

Response: Tree growth, amount of litterfall, and decomposition are intimately linked. On the one hand, tree productivity enhances the amount of litterfall and litter decomposition (Beugnon, Bu et al. 2023; Beugnon, Eisenhauer et al. 2023; Liu et al. 2018); on the other hand, increasing the amount of litterfall and decomposition enhances nutrient availability in soil and thereby tree growth (Freschet et al. 2013). Therefore, integrating tree biomass into the simulation might lead to a certain circularity by indirectly integrating plant-soil feedback mechanisms, and we wanted to provide a conservative measure of biodiversity effects rather than an overestimation. To address this issue, temporal data and models would be needed. We added explanatory text accordingly to the Discussion. Our approach first focuses on the relationship between the spatial organization of tree species and litterfall dynamics. We

therefore reduced interacting factors potentially blurring or inflating our results, by keeping covariates, such as tree biomass, constant.

L. 325

“Soil carbon storage depends on the decomposition and the amount of litter set to decompose^{9–11}. We fixed tree species biomass across the simulation scenarios; and thereby, the amount of litter falling on the ground at the forest level (Fig. 2), to avoid any potential nutrient feedback loops³⁵ and an overestimation of the explored effects. However, recent analyses from the same experiment showed that tree pairwise interactions underlie the biodiversity-ecosystem functioning relationship³⁶. In particular, tree productivity increases with an increasing proportion of inter-specific neighbours, by enhancing tree species spatial heterogeneity, i.e. an enhanced proportion of inter-specific interactions, increases tree productivity and biomass. Taking together Yu and colleagues³⁶ as well as our results highlights that increasing the spatial heterogeneity of tree species increases forest productivity, litterfall and litter decomposition for a given species richness of forest stand. Thus, improving tree productivity and stabilizing litter decomposition by increasing tree species spatial heterogeneity might become a powerful tool to optimize biodiversity benefits for nutrient recycling, carbon sequestration, and climate change mitigation.”

Beugnon, R., Bu, W., Bruelheide, H., Davrinche, A., Du, J., Haider, S., Kunz, M., von Oheimb, G., Perles-Garcia, M. D., Saadani, M., Scholten, T., Seitz, S., Singavarapu, B., Trogisch, S., Wang, Y., Wubet, T., Xue, K., Yang, B., Cesarz, S., & Eisenhauer, N. (2023). Abiotic and biotic drivers of tree trait effects on soil microbial biomass and soil carbon concentration. *Ecological Monographs*, 93(2), e1563. <https://doi.org/10.1002/ecm.1563>

Freschet, G. T., Cornwell, W. K., Wardle, D. A., Elumeeva, T. G., Liu, W., Jackson, B. G., Onipchenko, V. G., Soudzilovskaia, N. A., Tao, J., & Cornelissen, J. H. C. (2013). Linking litter decomposition of above- and below-ground organs to plant–soil feedbacks worldwide. *Journal of Ecology*, 101(4), 943–952. <https://doi.org/10.1111/1365-2745.12092>

Liu, X., Trogisch, S., He, J.-S., Niklaus, P. A., Bruelheide, H., Tang, Z., Erfmeier, A., Scherer-Lorenzen, M., Pietsch, K. A., Yang, B., Kühn, P., Scholten, T., Huang, Y., Wang, C., Staab, M., Leppert, K. N., Wirth, C., Schmid, B., & Ma, K. (2018). Tree species richness increases ecosystem carbon storage in subtropical forests. *Proceedings. Biological Sciences*, 285(1885). <https://doi.org/10.1098/rspb.2018.1240>

Ray, T., Delory, B. M., Beugnon, R., Bruelheide, H., Cesarz, S., Eisenhauer, N., Ferlian, O., Quosh, J., Von Oheimb, G., & Fichtner, A. (2023). Tree diversity increases productivity through enhancing structural complexity across mycorrhizal types. *Science Advances*, 9(40), eadi2362. <https://doi.org/10.1126/sciadv.adi2362>

Yu, W., Albert, G., Rosenbaum, B., Schnabel, F., Bruelheide, H., Connolly, J., Härdtle, W., Von Oheimb, G., Trogisch, S., Rüger, N., & Brose, U. (2024). Systematic distributions of interaction strengths across tree interaction networks yield positive diversity–productivity relationships. *Ecology Letters*, 27(1), e14338. <https://doi.org/10.1111/ele.14338>

Furthermore, it is not clear how the tree pair identities and interactions with surrounding trees beyond the TSP’s are included in the simulations and how they affect the results.

Response: We used a distance-based framework. Thus, all trees are included proportionally to their distance to the focal point. The TSP design ensured a good balance between focal species and micro-environments in the experimental dataset, and thus in the overall modeling. We revised the entire Methods section to increase clarity and added Supplementary Table 1 describing the selected species' functional traits.

The methodological presentation needs refinement for the broader audience of this journal, addressing concerns about clarity and assumptions.

Response: Agreed. We revised the entire Methods section to increase clarity.

Furthermore, the study's focus on litterfall and decomposition under the carbon sequestration context requires clarification. Carbon sequestration depends on the net balance between inputs and outputs, not just decomposition rates, a nuance overlooked in the study.

Response: We concur with the reviewer's observation that soil carbon sequestration stands in the balance between soil carbon inputs and outputs, positioning litter decomposition at a critical position. Previous studies from the same experiment have shown that tree species richness increased productivity litterfall, especially, neighborhood species-richness (Beugnon, Bu et al. 2023; Ray et al. 2023; Liu et al 2018), with consequences for soil carbon concentrations (Beugnon, Bu et al 2023; Liu et al. 2018). This positive effect of neighboring tree diversity on forest productivity has recently been linked to positive tree-tree interactions between heterospecific species pairs (Yu et al 2024). Therefore, increasing tree species spatial heterogeneity within the forest is expected to increase species-specific biomass, and thereby decomposition and carbon sequestration.

Beugnon, R., Bu, W., Bruelheide, H., Davrinche, A., Du, J., Haider, S., Kunz, M., von Oheimb, G., Perles-Garcia, M. D., Saadani, M., Scholten, T., Seitz, S., Singavarapu, B., Trogisch, S., Wang, Y., Wubet, T., Xue, K., Yang, B., Cesarz, S., & Eisenhauer, N. (2023). Abiotic and biotic drivers of tree trait effects on soil microbial biomass and soil carbon concentration. *Ecological Monographs*, 93(2), e1563. <https://doi.org/10.1002/ecm.1563>

Liu, X., Trogisch, S., He, J.-S., Niklaus, P. A., Bruelheide, H., Tang, Z., Erfmeier, A., Scherer-Lorenzen, M., Pietsch, K. A., Yang, B., Kühn, P., Scholten, T., Huang, Y., Wang, C., Staab, M., Leppert, K. N., Wirth, C., Schmid, B., & Ma, K. (2018). Tree species richness increases ecosystem carbon storage in subtropical forests. *Proceedings. Biological Sciences*, 285(1885). <https://doi.org/10.1098/rspb.2018.1240>

Ray, T., Delory, B. M., Beugnon, R., Bruelheide, H., Cesarz, S., Eisenhauer, N., Ferlian, O., Quosh, J., Von Oheimb, G., & Fichtner, A. (2023). Tree diversity increases productivity through enhancing structural complexity across mycorrhizal types. *Science Advances*, 9(40), eadi2362. <https://doi.org/10.1126/sciadv.adi2362>

Yu, W., Albert, G., Rosenbaum, B., Schnabel, F., Bruelheide, H., Connolly, J., Härdtle, W., Von Oheimb, G., Trogisch, S., Rüger, N., & Brose, U. (2024). Systematic distributions of

interaction strengths across tree interaction networks yield positive diversity–productivity relationships. *Ecology Letters*, 27(1), e14338. <https://doi.org/10.1111/ele.14338>

Additionally (a minor issue), clarification on the seasonality of litterfall measurements and whether all species shed leaves simultaneously (in December) would enhance the study's context and interpretation.

Response: Thank you for noticing this important point. The litterfall sampling period (September – December) covered the major litterfall season in this experiment, as measured by Huang and colleagues (2017) on the same species in this area. We added this information to the Methods section.

Huang, Y., Ma, Y., Zhao, K., Niklaus, P. A., Schmid, B., & He, J.-S. (2017). Positive effects of tree species diversity on litterfall quantity and quality along a secondary successional chronosequence in a subtropical forest. *Journal of Plant Ecology*, 10(1), 28–35. <https://doi.org/10.1093/jpe/rtw115>

In conclusion, while the study offers valuable insights, improvements in methodological clarity, addressing oversimplified assumptions, and bridging gaps between experimental and simulated conditions are necessary for a more robust and widely applicable contribution to the field.

Response: Thank you again for the helpful comments that helped addressing these outlined previous shortcomings.

Point-by-point response to reviewers' comments

Dear reviewers,

we would like to thank you for your time and positive feedback on our manuscript. We are grateful for the detailed and constructive suggestions provided that helped us to further improve our manuscript. Please find below a point-by-point response on how we addressed your comments. Please note that we have highlighted all reviewers' comments in *blue* font for the purpose of clarity. Line numbers cited below refer to the final document line numbers.

REVIEWER COMMENTS

Reviewer #1 (Remarks to the Author):

I found the authors have addressed the concerns from my previous review reports. I think the new analyses conducted on tree biomass strengthen the conclusion of this study. This study can notify researchers of the importance of considering the spatial heterogeneity in the BEF studies.

Thank you for your positive feedback and your comments to further increase the quality of our work. We adapted the text accordingly to solve the remaining issues.

One thing I'm newly concerned about is the description of the results related to Fig. 3. I think the authors can say that the effects of species richness on response variables were affected by spatial heterogeneity only when the interaction between tree species richness and tree species spatial heterogeneity was significant. The description of the results related to Fig. 3 was sometimes imprecise. So please describe those results more accurately.

We appreciate this comment and agree that our description of the figure was too concise; we expanded the results section accordingly and we provide more details to enhance clarity.

L. 202: "In order to evaluate how varying spatial heterogeneity modulates tree species richness effects on litterfall and decomposition, we repeated our simulations for two- and four-species mixtures and compared those to the results from the eight-species mixtures. Tree species richness increased tree biomass, the amount of litterfall, litter species richness, and litter decomposition, as well as, their spatial heterogeneity. Likewise, tree spatial heterogeneity increased litter species richness and decomposition rate but reduced litterfall and decomposition spatial heterogeneity (Fig. 3, see black arrows). Across the response variables, we showed a significant interaction effect between tree species richness and spatial heterogeneity ($p < 0.001$ except for tree biomass and the amount of litterfall, Fig. 3). We generally observed an increasing strength of forest stand tree species richness effects on the litterfall and decomposition response variables with increasing tree species spatial heterogeneity (Fig. 3.B-C); however, we did not identify such an effect on tree biomass variables (Fig. 3.A). For instance, carbon decomposition rate increased with tree species richness,

and this relationship strengthened with increasing tree species spatial heterogeneity ($p_{\text{interaction}} < 0.001$ and $R^2 = 54\%$, Fig. 3.C); thus, spatial heterogeneity boosted the effects of diversity on ecosystem functions. For block designs, tree species richness effects on decomposition rate became neglectable ($p > 0.05$, from 35.1 to 36.5% between 2- and 8-species mixtures, see Fig. 3.C), while in the random settings, they were more pronounced ($p < 0.001$, from 36.7 to 47.1%). In summary, our results show that the strength of the BEF relationship is contingent upon the spatial heterogeneity of tree species within the forest.”

In addition, I have listed some minor points that should be addressed below. I hope these comments will improve the manuscript.

Line 107: “litter composition” may be better as “litter species composition”.

Thanks for noticing; the text was corrected accordingly.

Line 157 & 159: “lows” should be “lines”

The text was modified accordingly.

Line 165: Please insert (Fig. 2A) after “designs”.

Thank you for this suggestion; the reference to the figure was added.

Line 168: Please insert (Fig. 2B) after “forest stand level”.

The text was modified accordingly.

Line 171: “2A” should be “2B”.

Thanks for noticing, the text was corrected accordingly.

In the Figure 2, in the illustration of “Plantation designs”, the order of “mini-blocks” and “double lines” should be replaced (i.e, the order should be Blocks, Mini-blocks, 2 lines, lines, and Random from left to right).

Thank you for pointing this out, the text was corrected accordingly.

Line 191 & 194: “2B” should be “2C”.

Thanks for noticing, the text was corrected accordingly.

Lines 202-204: I think this sentence would be somewhat misleading. You can say that tree species heterogeneity strengthens the effects of forest stand tree species richness on the response variables when the interaction between species richness (s) and tree species spatial heterogeneity (h) is significant. Among the 8 response variables in Fig. 3, 5 response variables show the significant interactions, but the variables related to tree biomass and litterfall, which are two of the most important variables in this study, show no significant interactions. Please describe the results more precisely.

We agree that our phrasing was not precise enough and revised the section accordingly.

L. 203: “In order to evaluate how varying spatial heterogeneity modulates tree species richness effects on litterfall and decomposition, we repeated our simulations for two- and four-species mixtures and compared those to the results from the eight-species

mixtures. Tree species richness increased tree biomass, the amount of litterfall, litter species richness, and litter decomposition, as well as, their spatial heterogeneity. Likewise, tree spatial heterogeneity increased litter species richness and decomposition rate but reduced litterfall and decomposition spatial heterogeneity (Fig. 3, see black arrows). Across the response variables, we showed a significant interaction effect between tree species richness and spatial heterogeneity ($p < 0.001$ except for tree biomass and the amount of litterfall, Fig. 3). We generally observed an increasing strength of forest stand tree species richness effects on the litterfall and decomposition response variables with increasing tree species spatial heterogeneity (Fig. 3.B-C); however, we did not identify such an effect on tree biomass variables (Fig. 3.A). For instance, carbon decomposition rate increased with tree species richness, and this relationship strengthened with increasing tree species spatial heterogeneity ($p_{\text{interaction}} < 0.001$ and $R^2 = 54\%$, Fig. 3.C); thus, spatial heterogeneity boosted the effects of diversity on ecosystem functions. For block designs, tree species richness effects on decomposition rate became neglectable ($p > 0.05$, from 35.1 to 36.5% between 2- and 8-species mixtures, see Fig. 3.C), while in the random settings, they were more pronounced ($p < 0.001$, from 36.7 to 47.1%). In summary, our results show that the strength of the BEF relationship is contingent upon the spatial heterogeneity of tree species within the forest.

”

Lines 211-212: Same comments as above. Is it only true for C decomposition rate?

Please see the revised section in the previous comments (L. 203)

Line 225: It's better to insert "distribution" after "litterfall", and "rate" after "litter decomposition".

Thank you for this suggestion; the text was modified accordingly.

Lines 229-230: This is not true for tree biomass at least?

We modified the sentence to better fit the results of our simulation.

L. 239 “The effects of tree species spatial heterogeneity on litterfall spatial heterogeneity and decomposition are particularly pronounced in species-rich forests (H2).”

Lines 268-271: I agree with this on decomposition rate, but not on biomass production.

We rephrased the sentence to improve its clarity.

L. 281: “Accordingly, the positive relationship between tree species richness and ecosystem functions, such as litterfall distribution and decomposition, is maximized when the different tree species are planted fully randomly and remains limited when the different tree species are aggregated into blocks, i.e. patches of single species (Fig. 3).”

Methods

Line 80: "litterfall and litter decomposition" should be "tree biomass, litterfall and litter decomposition"?

Thanks for noticing, the text was corrected.

Line 133: I was not able to find Suppl S4.

Thanks for noticing; the correct reference was Suppl. S2.

Suppl. S1

Page1: In “Plot designs”, “9-species mixture” should be “8-species mixture”.

Page 6: “9-species mixture” should be “8-species mixture”.

Page 6-8: “Small blocks” should be “Mini blocks”.

We apologize for these inconsistencies and thank you for noticing; the supplement was corrected accordingly.

Reviewer #2 (Remarks to the Author):

Response to authors

I have reviewed the revised manuscript and find that the authors have effectively addressed most of my previous concerns. However, I have a few additional and minor comments that I believe could further clarify and strengthen the paper.

We very much appreciate your positive feedback. We revised the text to solve the remaining issues.

Incorporation of Biomass Estimates

I appreciate the authors’ effort to integrate biomass estimates based on the methods from Yu et al. (2024). This addition enhances the analysis and is a valuable improvement to the manuscript. However, from the description in the Methods section, I am uncertain whether these modeled biomass values were integrated into the litterfall and decomposition simulations. From my understanding, biomass should ideally be recalculated in each simulation based on the specific pairwise interaction coefficients relevant to the species composition. This interpretation aligns with the main text and also seems logically consistent with the model framework. However, the methods section does not make this process fully clear. If biomass recalculations were indeed incorporated for each simulation, a brief mention in the methods section would clarify this point. A single sentence where relevant would be enough.

We thank the reviewer for this valuable comment. The biomass was predicted for each plot using the model from Yu et al. (2024) and the plot spatial configuration, which then informed the litterfall and decomposition predictions. We clarified this in the method section and Fig. 1.

Methods

L. 84: “Specifically, we used the output of the biomass simulation to predict litterfall, which then informed the litter decomposition model. The analyses were limited to the eight species matching the field sampling of litterfall and decomposition and the species predictable from the models by Yu and colleagues².”

L. 133: “Using this model fit, individual tree biomass was predicted for each simulated forest (i.e. species composition and spatial design); in particular, we set starting biomasses for all trees to 100 and simulated tree growth for 10 years for all the trees.”

Framing of Soil Carbon Storage

I still have some reservations about the framing of the study's introduction in terms of "soil carbon storage." The current study does not directly measure or model soil carbon storage. While tree biomass, litterfall, and decomposition processes are evaluated and the study concludes an increase in carbon storage within tree biomass, the impacts on soil carbon storage are not explicitly assessed here. Although it is reasonable to hypothesize a relationship with soil carbon (supported by existing literature), the decomposition and mass loss measures here do not directly inform the ultimate fate of carbon (e.g., storage vs. respiration).

We agree with the reviewer that our study does not include an explicit test of the fate of carbon during decomposition. The aim here in the Introduction was to refer to the existing evidence of how altered decomposition dynamics may affect soil C storage providing a general context. We reformulated the text to better account for the fact that this issue is not yet fully understood or resolved, especially in forest ecosystems and over longer time periods relevant for C storage. We believe that this more nuanced representation of the knowledge gap will inspire future research.

Definition of Spatial Heterogeneity

The definition of spatial heterogeneity still needs refinement to ensure replicability. Currently, it lacks sufficient detail for readers to fully replicate or understand its application in the study.

We added more details to the description in the main text and the methods.

L. 154: "We measured the spatial heterogeneity of tree species in a plot as the deviation of observed conspecific neighbours from those predicted by a null model (i.e., a randomly distributed eight-species mixture following a hypergeometric distribution)"

Method:

L. 107: "We measured the spatial heterogeneity of tree species in a plot as the deviation of observed conspecific neighbours from those predicted by a null model (i.e., a randomly distributed 8-species mixture). In a random plantation design, the number of heterospecific neighbors X follows a hypergeometric distribution, for which the expected value is $E[X_i] = \frac{n(K_i)}{T-1}$, where n defines the number of neighbouring trees, K_i is the number of trees that are not of the same species as tree i , and T is the total number of individual trees in the entire simulated plantation. Then, heterogeneity H was defined as $H = \sum(N(i) - E[X_i])$, where the variable i is iterating over all trees in the plantation, and $N(i)$ is the observed number of conspecific neighbours of tree i . For each plantation design, we calculated tree biomass, litterfall, and litter decomposition rates based on models fitted to empirical data."

Further, minor comments are provided in the attached files.

L. 73 Simultaneously increased litterfall and litter decomposition do not necessarily lead to higher carbon storage.

Yes, we agree with the reviewer. We modified the text accordingly. Please see also our response to a similar comment above.

L. 85: Higher decomposition can also translate into higher respiration and thus not in more soil carbon storage.

We agree that this question is not yet resolved and requires further testing. We reformulated this sentence to better account for the hypothetical nature of how decomposition rate is linked to soil C storage.

L. 77: “Spatial heterogeneity of tree species should maximize species interactions, leading to more evenly distributed litterfall that may facilitate decomposition⁹, thus enhancing soil carbon sequestration according to the microbial efficiency matrix stabilisation framework⁹ that, however, still remains to be tested in forest ecosystems and over longer time periods^{19,23}”

L. 177: Distance in terms of what= I think this metric is crucial and needs to be described better. I am not sure I could replicate this measurement with this description or the one in the methods section

Thank you for pointing out this shortcoming; we included more details in the description in the main text and in the methods.

L. 154: “We measured the spatial heterogeneity of tree species in a plot as the deviation of observed conspecific neighbours from those predicted by a null model (i.e., a randomly distributed eight-species mixture following a hypergeometric distribution).”

L. 200: In the plots the “2 lines” are not aligned with the label in the x axis. Also decomposition (%) and mass loss (g/cm²) are not rates, i.e. they are not per time unit.

Thank you for noticing the typos, these were corrected in the figure.

L. 436: Perhaps modify (or similar) instead of improve.

Thank you for the suggestion; the text was modified accordingly.

L. 334: “Simply transforming plantations of blocked species clusters into adjacent rows of different species was shown to already modify ecosystem functioning, with probably acceptable consequences for forest management.”

Methods

L. 32: Estimation is more appropriate.

Thanks for your comment; we adapted the wording to clarify.

L. 32: “Tree measurements and biomass estimations”

L. 89: Incorrect link.

We added the complete URL.

L. 78: “ All simulations, statistical analyses, and data visualizations were performed using R software version 4.3 (R Core Team¹³), and R-scripts are provided to the readers on our Zenodo directory (<https://doi.org/10.5281/zenodo.13808826>).”

L. 100: Not necessary

We modified the text accordingly.

L. 89: “From all the possible 2-, 4-, and 8-species mixtures combinations, we selected all 28 2-species mixture permutations (e.g. Sp1-Sp2, Sp2-Sp1 ...), and randomly chose 1,000 4- and 1,000 8-species mixtures permutations, totaling 2,028 species mixtures permutations.”

L. 124: Still find this definition unclear. Distance in terms of what? Please reformulate or add a mathematical formulation.

We have now detailed the description in the main text and methods.

L. 108: “We measured the spatial heterogeneity of tree species in a plot as the deviation of observed conspecific neighbours from those predicted by a null model (i.e., a randomly distributed eight-species mixture following a hypergeometric distribution).”